# Autoregressive Generative Adversarial Networks /
# Conference Submissions

## Abstract

Generative Adversarial Networks (GANs) learn a generative model by playing an adversarial game between a generator and an auxiliary discriminator, which classifies data samples vs. generated ones. However, it does not explicitly model feature co-occurrences in samples. In this paper, we propose a novel Autoregressive Generative Adversarial Network (ARGAN), that models the latent distribution of data using an autoregressive model, rather than relying on binary classification of samples into data/generated categories. In this way, feature co-occurrences in samples can be more efficiently captured. Our model was evaluated on two widely used datasets: CIFAR-10 and STL-10. Its performance is competitive with respect to other GAN models both quantitatively and qualitatively.

## 1 Introduction

Generative Adversarial Networks (GANs) (Goodfellow et al., 2014) are a new type of generative model with wide-spread success. For example, they can be used to generate photo-realistic face and bedroom images (Radford et al., 2015). Natural images contain a rich amount of features with specific co-occurrence configurations. A generative model should be able to learn these configurations in order to perform well. In a GAN setting, a discriminator is used to learn features in data and helps the generator to imitate them. The generator should be able to produce samples with certain feature configurations – which are seen as likely by the discriminator – to satisfy the discriminator's criteria.

However, GAN's performance does not scale well to large datasets with high sample variation such as ImageNet (Russakovsky et al., 2015). It can generate realistic textures, but the global consistency is still lacking (Salimans et al., 2016). Recently several variations of GAN have been proposed (Arjovsky et al., 2017; Mao et al., 2016; Zhao et al., 2016) to improve image generation. However the issue of global inconsistency still persists.

We postulate that the issue of global inconsistency has to do with the discriminator of a GAN and especially its classification nature to model detailed global structure. The discriminator architecture is similar to ones used in classification or recognition tasks, whose aim is modeling $p(y|x)$ but not $p(x)$, where $y$, $x$, $p(x)$ represent class probabilities, data sample and data distribution, respectively. However the overall aim of generative modeling is learning $p(x)$. A good "teacher" a.k.a. discriminator, should be able to learn $p(x)$ in order to teach it to its "student" a.k.a. generator, but the discriminator only learns $p(x)$ implicitly by modeling $p(y|x)$ in an adversarial way. However, direct modeling of $p(x)$ is time consuming and poses difficulties for capturing global structure in high resolution images (van den Oord et al., 2016).

In order to alleviate this problem, we propose a latent space modeling where an adversarial encoder learns the high dimensional feature vectors of real samples that are distinguishable from the fake ones. The distinguishability criteria is the modeling error of an autoregressive model that tries to model latent space distribution of real images $p(f)$ where $f$ represents high-dimensional features. A good adversarial encoder should come up with features that produce low error if they belong to real images and high error if they belong to fake ones. Different from the basic GAN, $p(y|x)$, discriminator, is not modeled but rather $p(f|x)$, the encoder, and $p(f)$, the autoregressive model.

Our proposed Autoregressive GAN (ARGAN) learns features in the real data distribution without classification but instead by modeling the latent space of the encoder with an autoregressive model. Since an autoregressive model factorizes latent space and learns features in a conditional way, it can model feature co-occurrences more effectively than a classifier as used in conventional GAN.

Furthermore, we propose a combination of ARGAN and Patch-GAN (Isola et al., 2016). In this setting, Patch-GAN ensures realistic local features while ARGAN deploys its capacity at the global level. We show that PARGAN improves both qualitative and quantitative results of the generated images.

We will release the source code and model data upon paper acceptance.

## 2 RELATED WORK

In various GAN models, the discriminator can be partitioned into two parts, a feature learner (encoder) and an assigner, which assigns values to distinguish two distributions either data or model. A vanilla GAN learns features by using an adversarial encoder (E) and a linear classifier (c) as assigner (Fig. 1). The Wasserstein GAN (WGAN) (Arjovsky et al., 2017) follows a similar idea and uses an adversarial encoder with a linear separator (s) as assigner, which separates real features from generated ones on a single dimensional space. Energy-based GAN (EBGAN) (Zhao et al., 2016) slightly departs from this trend and uses an encoder to learn features and a decoder (D) as assigner, which reconstructs input of encoder. In EBGAN, different from the previous methods, energy value is not the output of the assigner but the reconstruction. In the proposed ARGAN model, features are learned with an encoder, and an autoregressive model (R) is used as assigner. Similar to EBGAN, energy value is not the assigner's output but rather its difference from the target, which corresponds to the modeling error of the assigner. For the generator all models are the same, a noise sample (z) is fed into a generator (G) that maps it into the data space.

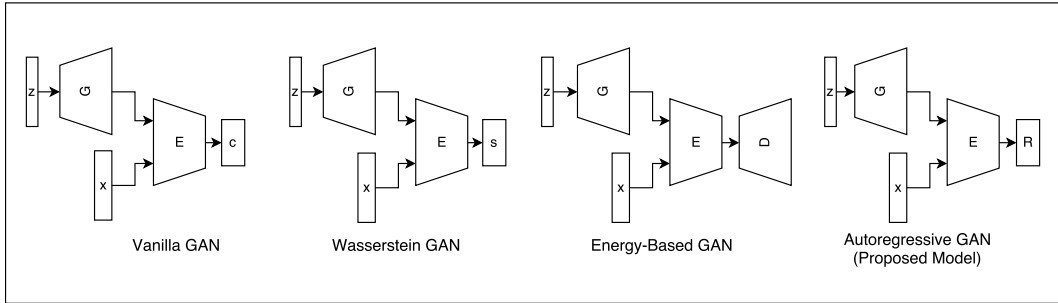

Figure 1: A simplified comparison of GAN models including the proposed Autoregressive GAN.

Another type of generative model is proposed by van den Oord et al. (2016) and Theis & Bethge (2015), who model the joint distribution of pixels, $p(x)$, with an RNN in an autoregressive way. During test time samples are generated by sampling one pixel at a time from the model. In our work, autoregressive modeling is used in the latent space, and the features incoming to autoregressor are learned in an adversarial manner. Hence instead of modeling data in the data space, they are modeled in the feature space, which is learned by an adversarial encoder concurrently. Modeling distribution in the latent space can focus on semantics of data rather than low level features like correlation between adjacent pixels. Furthermore, our model runs as fast as vanilla GAN during test time since recurrent modeling is only used during training as an auxiliary model.

The closest work to our method is Denoising Feature Matching Warde-Farley & Bengio (2017). Even though Denoising Feature Matching uses density estimation in the latent space, there are major differences which makes learning dynamic of our model totally different then theirs. (i) Their method is complementary to GAN objective while our method can be learned standalone. (ii) More importantly their discriminator (encoder + classifier) are trained as in original GAN objective which means that features learned from the data distribution are based on classifier's feedback not on density model's. This crucial difference make both works different than one another. (iii) In our model feature co-occurrences is modeled explicitly. (iv) Motivation for both works are totally different.

## 3 METHOD

### 3.1 PRELIMINARY

GAN is a two player min-max game between a generator $G$ and a discriminator $D$, as formulated by Eq. 1. $G$ takes a noise sample $z$ and maps it into data space $G(z; \theta_g)$, while $D$ tries to discriminate between samples coming from $G$ and real samples $x$. Overal, $D$'s aim is to learn features that separate samples coming from $p_g$, generator's distribution, and samples coming from $p_{data}$, real data distribution. $G$'s aim is to produce samples that are indistinguishable from the real ones. In practice, the objective for $G$ is changed with Eq. 3 in order to improve gradient flow, especially during the initial part of the training. At each iteration, $D$ maximizes Eq. 2 while $G$ maximizes Eq. 3. This procedure continues until it reaches the equilibrium $p_g \approx p_{data}$, which is a saddle point of Eq. 1.

$$\min_G \max_D E_{x \sim p_{data}(x)}[\log D(x)] + E_{z \sim p_z(z)}[\log(1 - D(G(z)))] \tag{1}$$

$$\max_D E_{x \sim p_{data}(x)}[\log D(x)] + E_{z \sim p_z(z)}[\log(1 - D(G(z)))] \tag{2}$$

$$\max_G E_{z \sim p_z(z)}[\log(D(G(z)))] \tag{3}$$

### 3.2 AUTOREGRESSIVE GAN (ARGAN)

The discriminator of GAN can be separated into two parts: $D = d \circ E$ where $E$ is an adversarial encoder, and $d$ is a linear binary classifier. The objective of $E$ is to produce linearly separable distributions of features $f_{real} \sim E(x)$ and $f_{fake} \sim E(G(z))$. Instead of using a classifier $d$ to separate distributions, we propose to use an autoregressive model, $R$, to model the latent distribution of real samples $p(f_{real})$. This model should produce a small error for samples coming from $p(f_{real})$ and a large error for samples coming from $p(f_{fake})$ so that it can differentiate one distribution from the other.

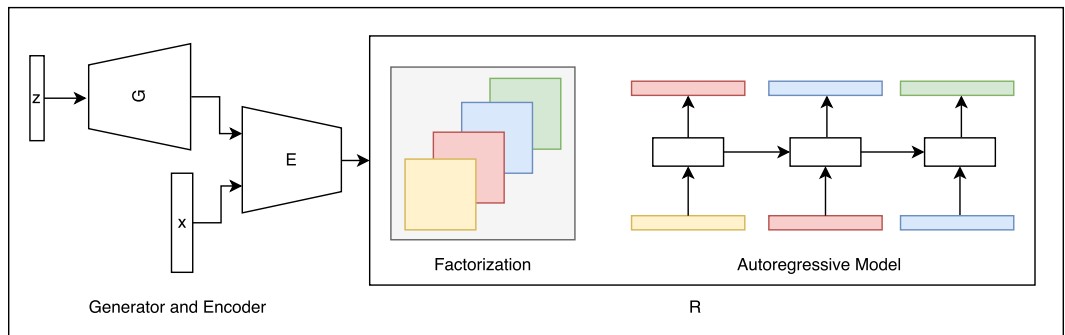

Figure 2: Proposed Autoregressive GAN (ARGAN) architecture.

An overview of the proposed ARGAN architecture is depicted in Fig. 2. When compared to vanilla GAN (Fig. 1), this model replaces the classifier layer with a factorization of features and autoregressive modeling of the factorized features. The autoregressive model can be represented with a multi-layer LSTM or other types of RNN models. Instead of using class discrimination as the learning signal, the autoregressive model's prediction loss is used. The intuition is that an autoregressive model would model the feature distribution better than fully connected layers by factorizing $f$ and modeling conditional distributions:

$$p(f) = p(f_1, ..., f_N) = \prod_{n=1}^{N} p(f_n | f_{<n}). \tag{4}$$

The proposed Autoregressive GAN (ARGAN) is formulated as in Eq. 5, where $G$ is the generator function, $E$ is the encoder, $\mathcal{L}$ is the loss function as defined in Eq. 6, and $\theta_G$, $\theta_E$, $\theta_R$ are the parameters for generator, encoder, and autoregressive model, respectively.

$$\min_{\theta_G} \max_{\theta_E, \theta_R} E_{x \sim p_{data}(x)}[\mathcal{L}(E(x; \theta_E), \theta_R)] - E_{z \sim p_z}[\mathcal{L}(E(G(z; \theta_G); \theta_E); \theta_R)] \tag{5}$$

$$\mathcal{L}(f; \theta_R) = \sum_{i=1}^{N_f} \log p_{\theta_R}(f_i | f_{<i}) \tag{6}$$

In practice, we found that Eq. 5 is unstable due to $\theta_R$'s maximization of the second term being unbounded. Instead $\theta_R$ is optimized only over the first term of Eq. 5. In this case, the objective for $G$ and $E$ stays the same (Eq. 7), while $\theta_R$ is optimized only on real sample features that are coming from $E(x)$ but not on $E(G(z))$ (Eq. 8):

$$\min_{\theta_G} \max_{\theta_E} E_{x \sim p_{data}(x)}[\mathcal{L}(E(x; \theta_E); \theta_R)] - E_{z \sim p_z}[\mathcal{L}(E(G(z; \theta_G); \theta_E); \theta_R)] \tag{7}$$

$$\max_{\theta_R} E_{x \sim p_{data}(x)}[\mathcal{L}(E(x; \theta_E); \theta_R)] \tag{8}$$

When $\theta_E$ maximizes Eq. 7, it tries to learns features that yield low error for real samples and high error for generated ones. In this way, $E$ exploits structures that are common on data samples but not on generated ones. When $\theta_R$ maximizes Eq. 8, it fits to features coming from real examples only.

We have not used any distribution function at the output layer of $R$ since it is not meant to be sampled but rather used during training as an auxiliary model. Using the model only during training removes test time sampling error (Huszár, 2015), which happens when conditional samples come from model distribution rather than data distribution.

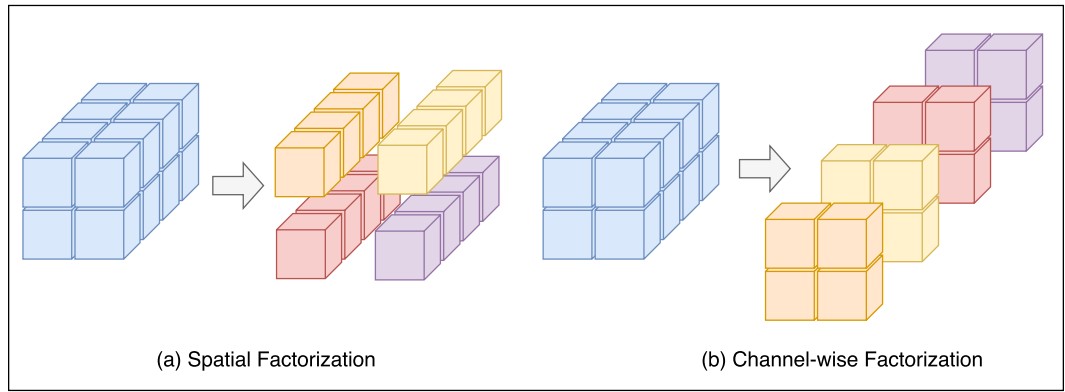

(a) Spatial Factorization                    (b) Channel-wise Factorization

Figure 3: (a) Spatial Factorization: $2 \times 2 \times 4$ feature tensor factorized into $1 \times 1 \times 4$ tensors which corresponds to factorization along spatial dimensions, (b) Channel-wise Factorization: $2 \times 2 \times 4$ feature tensor factorized into $2 \times 2 \times 1$ tensors which corresponds to factorization along depth (or channel) dimensions.

We do make the independence assumption when factorizing features in order to reduce computation time during training. Instead of conditioning single features on another, feature groups are conditioned on other feature groups by making the assumption that features in the same groups are independent from each other. Fig. 3 shows two types of factorization we use to model Spatial-wise ARGAN (S-ARGAN) and Channel-wise ARGAN (C-ARGAN). Even though the independence assumption does not strictly hold, in practice both of them work equally well.

### 3.3 PATCH-GAN + S-ARGAN (PARGAN)

Unlike vanilla GAN which produces single value for each sample, Patch-GAN (Isola et al., 2016), (Li & Wand, 2016) takes multiple patches from a sample and produces a value for each patch. Hence for a single sample, multiple scores from different regions are evaluated. Even though Patch-GAN can help the generator to produce locally realistic samples, it does not take the global arrangement into account. Isola et al. (2016) used Patch-GAN combined with L1 loss for image-to-image translation. While Patch-GAN ensures realism of local content, a pixel-wise loss with ground truth is used to enforce global arrangement.

For image generation as in our work, there is no ground truth to relate with $G(z)$, hence it is not clear how to enforce global arrangement. Therefore, in this paper, we propose S-ARGAN to ensure global

---

**Algorithm 1** ARGAN Training Procedure

---

1: $\theta_G, \theta_E, \theta_R \leftarrow$ initialize network parameters
2: **repeat**
3: $\quad x^{(1)}, ..., x^{(M)} \sim p_{data}(x)$ $\qquad\qquad\qquad\qquad$ ▷ Sample from data distribution
4: $\quad z^{(1)}, ..., z^{(M)} \sim p_{noise}(z)$ $\qquad\qquad\qquad\qquad$ ▷ Sample from noise distribution
5: $\quad f_r^{(j)} \leftarrow E(x^{(j)})$, for j = 1,...,M $\qquad\qquad$ ▷ Compute features for training examples
6: $\quad f_f^{(j)} \leftarrow E(G(z^{(j)}))$, for j = 1,...,M $\qquad$ ▷ Compute features for generated samples
7: $\quad p_r^{(j)} \leftarrow R(f_r^{(j)})$, for j = 1,...,M $\qquad\qquad$ ▷ Compute predictions for training examples
8: $\quad p_f^{(j)} \leftarrow R(f_f^{(j)})$, for j = 1,...,M $\qquad\qquad$ ▷ Compute predictions for training examples
9: $\quad \mathcal{L}_{\theta_G} \leftarrow \frac{1}{M}\sum_{j=1}^M \sum_{i=1}^{N_f} |p_{f_i}^{(j)} - f_{f_{i+1}}^{(j)}|$ $\qquad\qquad$ ▷ Compute loss for generator
10: $\quad \mathcal{L}_{\theta_E} \leftarrow \frac{1}{M}\sum_{j=1}^M \sum_{i=1}^{N_f} |p_{r_i}^{(j)} - f_{r_{i+1}}^{(j)}| - \frac{1}{M}\sum_{j=1}^M \sum_{i=1}^{N_f} |p_{f_i}^{(j)} - f_{f_{i+1}}^{(j)}|$
11: $\quad \mathcal{L}_{\theta_R} \leftarrow \frac{1}{M}\sum_{j=1}^M \sum_{i=1}^{N_f} |p_{r_i}^{(j)} - f_{r_{i+1}}^{(j)}|$ $\qquad$ ▷ Compute loss for autoregressive model
12: $\quad \theta_G \leftarrow \theta_G - \nabla_{\theta_G}\mathcal{L}_{\theta_G}$ $\qquad\qquad\qquad$ ▷ Update parameters of generator
13: $\quad \theta_E \leftarrow \theta_E - \nabla_{\theta_E}\mathcal{L}_{\theta_E}$ $\qquad\qquad\qquad$ ▷ Update parameters of encoder
14: $\quad \theta_R \leftarrow \theta_R - \nabla_{\theta_R}\mathcal{L}_{\theta_R}$ $\qquad\qquad$ ▷ Update parameters of autoregressive model
15: **until** convergence

---

consistency. In this case, Patch-GAN decides whether individual patches are realistic or not, while S-ARGAN examines whether their arrangement is spatially consistent. Beware that S-ARGAN by itself can give feedback into individual channels, however using it with Patch-GAN makes training easier and produces better results. Fig. 4 shows how the latent space of the encoder is factorized and feeds into S-ARGAN and Patch-GAN.

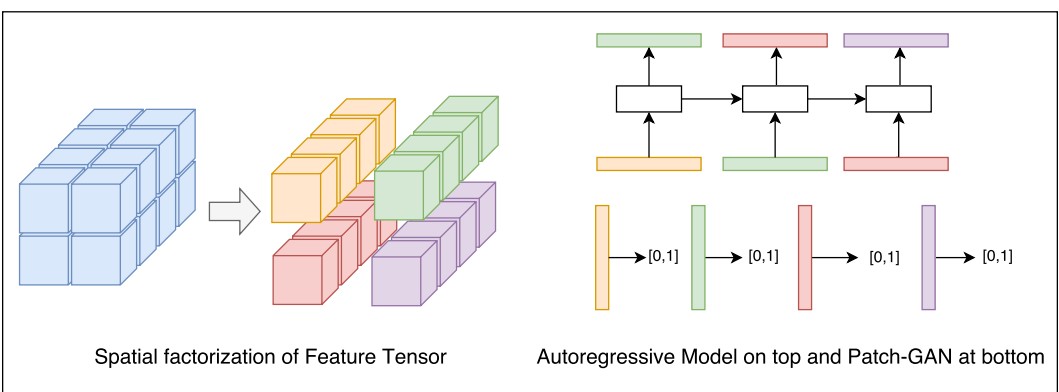

Spatial factorization of Feature Tensor $\qquad\qquad$ Autoregressive Model on top and Patch-GAN at bottom

Figure 4: Left: A feature tensor factorized in spatial fashion. Right (top): An autoregressive model on factorized features. Right (bottom): Patch-GAN: a single discriminator is used for each spatial tensor

The objective of the Patch-GAN for discriminator and generator is given in Eq. 9 and Eq. 10 where $D = d \oplus E$, $\oplus$ is convolution operation, $E(.) \in \mathbb{R}^{h \times w \times c}$ is the same encoder used in previous section and $d \in \mathbb{R}^{c \times 1 \times 1 \times 1}$ is the convolutional kernel. $D(.)$ produces an activation map with size $\mathbb{R}^{h \times w}$. Each activation is matched to its corresponding target by using least square loss. We use the least square version of GAN objective (Mao et al., 2016) rather than vanilla GAN objective (Eq. 2, Eq. 3) since vanilla GAN produces high error for generator later in the training. PARGAN objective is a simple summation of ARGAN objective (Eq. 7), Eq. 8 and Patch-GAN objective (Eq. 9), Eq. 10) without any hyperparameter.

$$\max_D E_{x \sim p_{data}(x)}[\frac{1}{hw}\sum_h \sum_w (D(x)_{h,w} - 1)^2] + E_{z \sim p_z(z)}[\frac{1}{hw}\sum_h \sum_w (D(G(z))_{h,w})^2] \qquad (9)$$

$$\max_G E_{z \sim p_z(z)}[\frac{1}{hw}\sum_h \sum_w (D(G(z))_{h,w} - 1)^2] \qquad (10)$$

## 4 EXPERIMENTS

We have conducted experiments on CIFAR-10 (Krizhevsky et al.), STL-10 (Coates et al., 2011) and CelebA (Liu et al., 2015) databases. CIFAR-10 has 60,000 images of $32 \times 32$ pixels with 10 classes: airplane, automobile, bird, cat, dear, dog, frog, horse, ship, and truck. We have used 50,000 training set images to train our model. STL-10 is a subset of ImageNet and more complex than CIFAR-10. There are 13,000 labeled and 100,000 unlabeled images with $96 \times 96$ pixels resolution. CelebA contains 200K aligned face images. The images are cropped to leave face are roughly. All of our experiments have been conducted without any labels from datasets.

There is no agreed-upon evaluation metric for generative models (Theis et al., 2015). One commonly used metric is the Inception score (Salimans et al., 2016) which is well correlated with human judgment. Inception score is $exp(E_x[KL(p(y|x)||p(y))])$, where $p(y|x)$ is the conditional class probability which should ideally have low entropy, and $p(y)$ is the marginal class probability which should ideally have high entropy. Higher values of inception score are better. For evaluation, 50,000 images are generated and fed into the official Inception score code[1] with 10 splits.[2]

The network architecture for each dataset is similar to Radford et al. (2015); details are provided in the Appendix. We have used transposed convolution with strides 2 in generator until it matches the size of the data. In the discriminator, convolutions with stride 2 are used until it reaches a pre-defined feature map size. BatchNorm (Ioffe & Szegedy, 2015) is used in both generator and discriminator, except the last layer of generator and first layer of discriminator. When C-ARGAN is used, the depth of the last layer in $E$ restricted to 256 in order to reduce computation time during training. In the case of S-ARGAN and PARGAN, the depth of the last layer is 512. We have used single layer LSTM for S-ARGAN, and multi-layer LSTM for C-ARGAN since single level produces blurry results. ADAM is used as optimizer with learning rate 0.0001 and $\beta_1 = 0.5$.

Interestingly, the L2 loss could not produce results as good as L1 loss in $R$'s objective (Eq. 6). Hence, we have conducted all our experiments using L1 loss in the autoregressive model. In the case of C-ARGAN and S-ARGAN, the generator is updated $t$ times to catch encoder. We found $t$ to be an important hyper-parameter to tune. By using grid search, we choose $t = 3$ in all our experiments. With this setting, the error of the generator closely follows the error of the encoder.

### 4.1 QUALITATIVE RESULTS

Generated images for the proposed S-ARGAN, C-ARGAN and PARGAN for CIFAR-10 are shown in Fig. 5. Note that none of the images in this paper are cherry picked. Overall images look sharp even though not all of them feature recognizable objects. For CIFAR-10, only 32x32 resolution is explored. Fig. 6 shows 48x48 pixel image generation from STL-10 dataset for S-ARGAN, C-ARGAN and PARGAN. We have also used PARGAN on the 96x96 STL-10 to show it can scale to higher resolutions (Fig. 7). Certain objects like car, ship, horse, and some other animals are recognizable. For CelebA generation we have used SW-ARGAN with 64x64 resolution (Fig. 8). The face images looks realistic in general with small artifacts in certain samples.

### 4.2 QUANTITATIVE RESULTS

Inception scores for the proposed S-ARGAN, C-ARGAN and PARGAN on CIFAR-10 and STL-10 are given in Table 1. We compare our results with other baseline GAN models[3] like DCGAN (Radford et al., 2015), EBGAN (Dai et al., 2017), WGAN (Gulrajani et al., 2017) rather than extensions such as D2GAN (Dinh Nguyen et al., 2017), MGGAN (Hoang et al., 2017), DFM (Warde-Farley & Bengio, 2017). These extensions can be combined with our model in the same way as they are combined with vanilla GAN. C-ARGAN and S-ARGAN show competitive results

---

[1] https://github.com/openai/improved-gan/tree/master/inception_score

[2] In other papers information about which model is used when reporting the score is often missing. For example, one can run the same model several times and publish the best score. In order to produce unbiased estimates, we have run our model and collected the score more than 10 times when there is no perceptual improvement over generated images. We published the averages of the scores.

[3] To be clear, each baseline GAN optimizes different divergence, for example basic GAN, WGAN, EGBAN, optimizes Jensen-Shannon, Wasserstein and total variance respectively. While extension does not change divergence but rather model architecture etc.

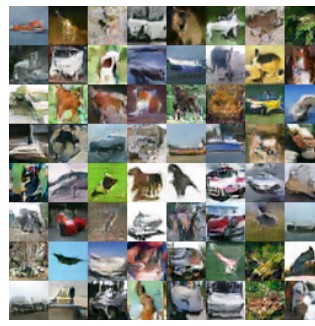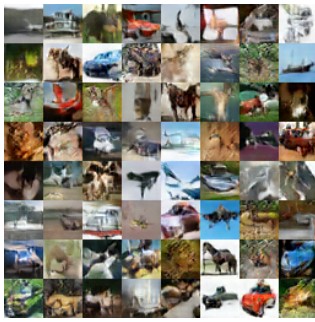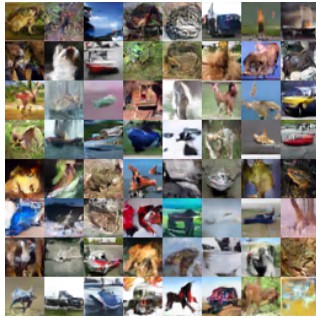

Figure 5: Generation for CIFAR-10 dataset. (Left): S-ARGAN. (Middle): C-ARGAN. (Right): PARGAN

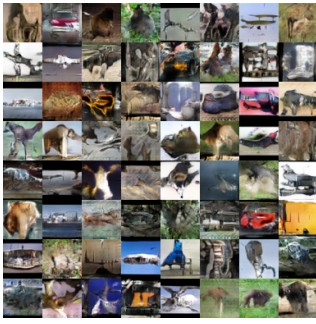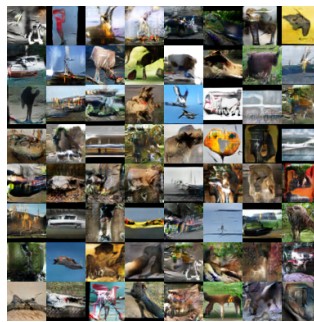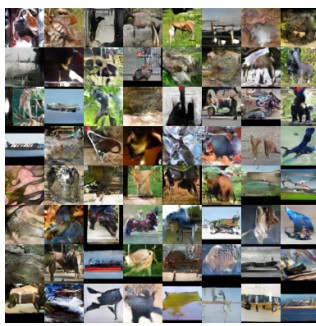

Figure 6: Generation for STL-10 dataset with 48x48 resolution. (Left): S-ARGAN. (Middle): C-ARGAN. (Right): PARGAN

with vanilla GAN, while PARGAN outperforms them. PARGAN is competitive with WGAN-GP on the same architecture (DCGAN) even though WGAN-GP performs better with ResNet (Gulrajani et al., 2017).

Table 1: Inception scores on CIFAR-10 and STL-10.

| Model | CIFAR-10 | STL-10 |
|---|---|---|
| Real data | $11.24 \pm 0.16$ | $26.08 \pm 0.26$ |
| ALI  (Dumoulin et al., 2016) | $5.34 \pm 0.05$ | - |
| BEGAN  (Berthelot et al., 2017) | 5.62 | - |
| D2GAN  (Dinh Nguyen et al., 2017) | $7.15 \pm 0.07$ | 7.98 |
| MGGAN  (Hoang et al., 2017) | 8.23 | 9.09 |
| DFM  (Warde-Farley & Bengio, 2017) | $7.72 \pm 0.13$ | $8.51 \pm 0.13$ |
| Improved-GAN  (Salimans et al., 2016) | $6.86 \pm 0.06$ | - |
| DCGAN  (Radford et al., 2015) | $6.40 \pm 0.05$ | 7.54 |
| EBGAN  (Zhao et al., 2016) | $6.74 \pm 0.09$ | - |
| WGAN-GP (ResNet)  (Gulrajani et al., 2017) | $7.86 \pm 0.07$ | $9.05 \pm 0.12$ |
| WGAN-GP (DCGAN) (Our implementation) | 6.80 | - |
| S-ARGAN (Proposed) | 6.50 | 7.44 |
| C-ARGAN (Proposed) | 6.46 | 7.60 |
| PARGAN (Proposed) | 6.86 | 7.89 |

## 5 CONCLUSIONS AND FUTURE WORK

We have proposed a novel GAN model and successfully trained it on CIFAR-10 and STL-10 datasets. It models the latent distribution of real samples, which is learned via adversarial training.

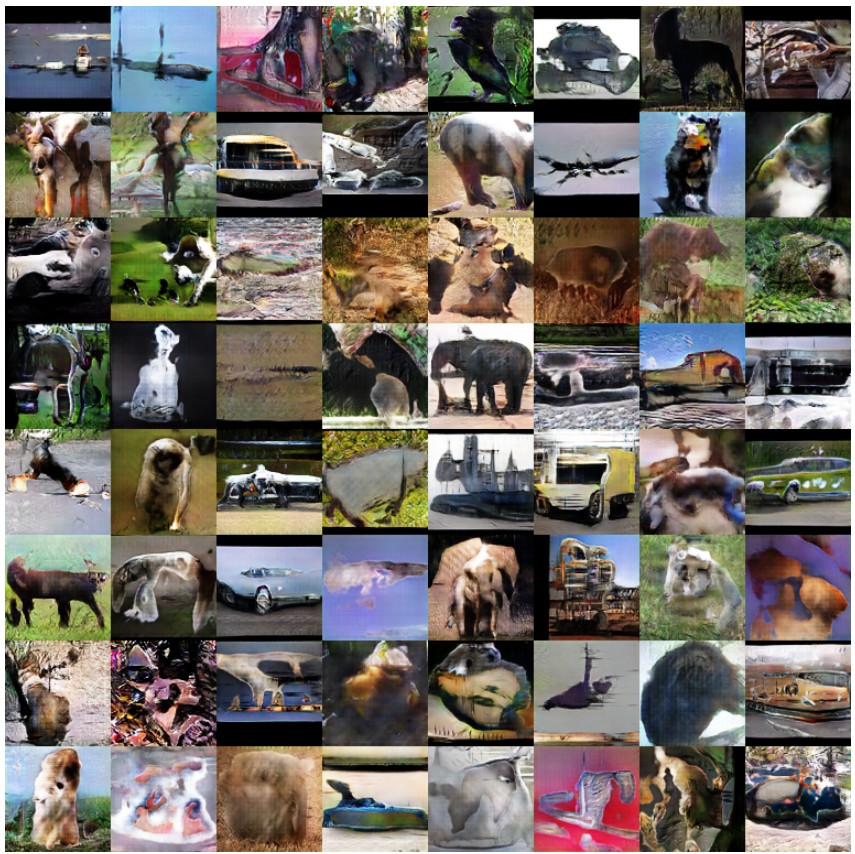

Figure 7: Generation for STL-10 dataset with 96x96 resolution PARGAN.

Besides, we have combined the proposed model with Patch-GAN to further improve the stability of training and the quality of generated images.

In this work, we have used a simple autoregressive model. Our model can be further enhanced by more advanced latent space modeling, such as bidirectional modules (Schuster & Paliwal, 1997), 2D LSTM (Theis & Bethge, 2015), hierarchical modules (Mehri et al., 2017) etc.

As future work, we would like to extend this method to ImageNet which has even larger sample variation. Combining our models with MGGAN (Hoang et al., 2017) is another potential research direction.

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

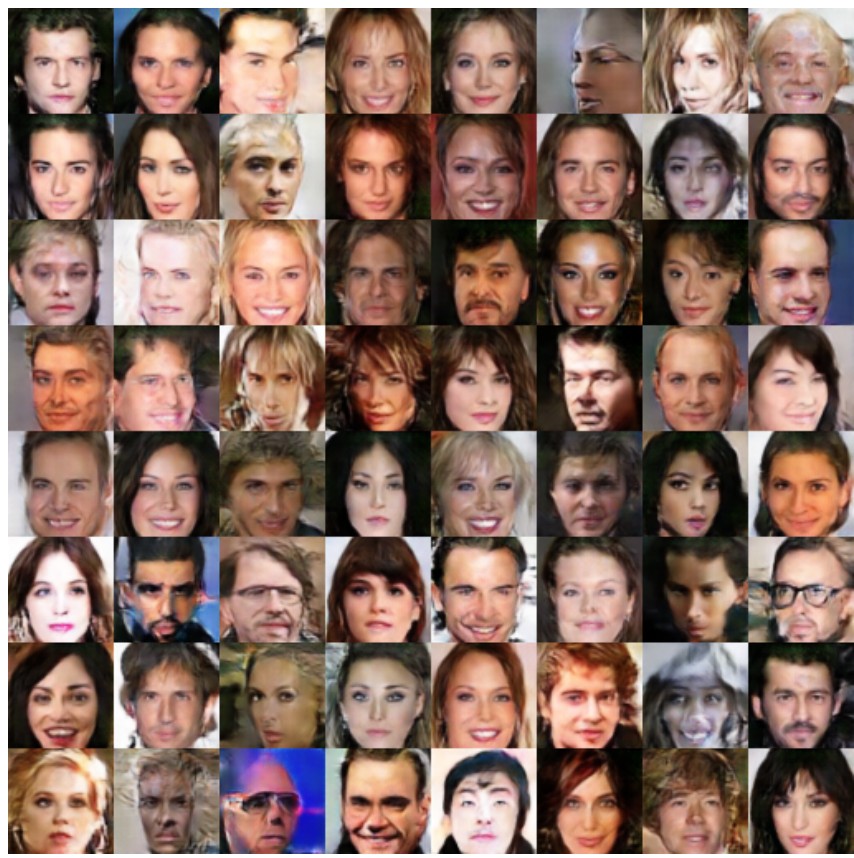

Figure 8: Generation for CelebA dataset with 64x64 resolution SW-ARGAN.

T. Dinh Nguyen, T. Le, H. Vu, and D. Phung. Dual Discriminator Generative Adversarial Nets. *ArXiv e-prints*, September 2017.

V. Dumoulin, I. Belghazi, B. Poole, O. Mastropietro, A. Lamb, M. Arjovsky, and A. Courville. Adversarially Learned Inference. *ArXiv e-prints*, June 2016.

Ian Goodfellow, Jean Pouget-Abadie, Mehdi Mirza, Bing Xu, David Warde-Farley, Sherjil Ozair, Aaron Courville, and Yoshua Bengio. Generative adversarial nets. In Z. Ghahramani, M. Welling, C. Cortes, N. D. Lawrence, and K. Q. Weinberger (eds.), *Advances in Neural Information Processing Systems 27*, pp. 2672–2680. Curran Associates, Inc., 2014. URL http://papers.nips.cc/paper/5423-generative-adversarial-nets.pdf.

I. Gulrajani, F. Ahmed, M. Arjovsky, V. Dumoulin, and A. Courville. Improved Training of Wasserstein GANs. *ArXiv e-prints*, March 2017.

Q. Hoang, T. Dinh Nguyen, T. Le, and D. Phung. Multi-Generator Generative Adversarial Nets. *ArXiv e-prints*, August 2017.

F. Huszár. How (not) to Train your Generative Model: Scheduled Sampling, Likelihood, Adversary? *ArXiv e-prints*, November 2015.

S. Ioffe and C. Szegedy. Batch Normalization: Accelerating Deep Network Training by Reducing Internal Covariate Shift. *ArXiv e-prints*, February 2015.

P. Isola, J.-Y. Zhu, T. Zhou, and A. A. Efros. Image-to-Image Translation with Conditional Adversarial Networks. *ArXiv e-prints*, November 2016.

Alex Krizhevsky, Vinod Nair, and Geoffrey Hinton. Cifar-10 (canadian institute for advanced research). URL http://www.cs.toronto.edu/~kriz/cifar.html.

C. Li and M. Wand. Precomputed Real-Time Texture Synthesis with Markovian Generative Adversarial Networks. *ArXiv e-prints*, April 2016.

Ziwei Liu, Ping Luo, Xiaogang Wang, and Xiaoou Tang. Deep learning face attributes in the wild. In *Proceedings of International Conference on Computer Vision (ICCV)*, 2015.

Xudong Mao, Qing Li, Haoran Xie, Raymond Y. K. Lau, and Zhen Wang. Multi-class generative adversarial networks with the L2 loss function. *CoRR*, abs/1611.04076, 2016. URL `http://arxiv.org/abs/1611.04076`.

Soroush Mehri, Kundan Kumar, Ishaan Gulrajani, Rithesh Kumar, Shubham Jain, Jose Sotelo, Aaron Courville, and Yoshua Bengio. Samplernn: An unconditional end-to-end neural audio generation model. 2017. URL `https://openreview.net/forum?id=SkxKPDv5xl`.

Alec Radford, Luke Metz, and Soumith Chintala. Unsupervised representation learning with deep convolutional generative adversarial networks. *CoRR*, abs/1511.06434, 2015. URL `http://arxiv.org/abs/1511.06434`.

Olga Russakovsky, Jia Deng, Hao Su, Jonathan Krause, Sanjeev Satheesh, Sean Ma, Zhiheng Huang, Andrej Karpathy, Aditya Khosla, Michael Bernstein, Alexander C. Berg, and Li Fei-Fei. ImageNet Large Scale Visual Recognition Challenge. *International Journal of Computer Vision (IJCV)*, 115(3):211–252, 2015. doi: 10.1007/s11263-015-0816-y.

Tim Salimans, Ian J. Goodfellow, Wojciech Zaremba, Vicki Cheung, Alec Radford, and Xi Chen. Improved techniques for training gans. *CoRR*, abs/1606.03498, 2016. URL `http://arxiv.org/abs/1606.03498`.

M. Schuster and K.K. Paliwal. Bidirectional recurrent neural networks. *Trans. Sig. Proc.*, 45(11): 2673–2681, November 1997. ISSN 1053-587X. doi: 10.1109/78.650093. URL `http://dx.doi.org/10.1109/78.650093`.

L. Theis and M. Bethge. Generative Image Modeling Using Spatial LSTMs. *ArXiv e-prints*, June 2015.

L. Theis, A. van den Oord, and M. Bethge. A note on the evaluation of generative models. *ArXiv e-prints*, November 2015.

Aäron van den Oord, Nal Kalchbrenner, and Koray Kavukcuoglu. Pixel recurrent neural networks. *CoRR*, abs/1601.06759, 2016. URL `http://arxiv.org/abs/1601.06759`.

David Warde-Farley and Yoshua Bengio. Improving generative adversarial networks with denoising feature matching. 2017. URL `https://openreview.net/forum?id=S1X7nhsxl`.

Junbo Jake Zhao, Michaël Mathieu, and Yann LeCun. Energy-based generative adversarial network. *CoRR*, abs/1609.03126, 2016. URL `http://arxiv.org/abs/1609.03126`.

## A NETWORK ARCHITECTURES

Network architecture for CIFAR-10:

- Generator: FC(100,512*4*4)-BN-ReLU-DC(512,256;4c2s)-BN-ReLU-DC(256,128;4c2s)-BN-ReLU-DC(128,3;4c2s)-Tanh
- Feature Learner: CV(3,64;4c2s)-BN-LRec-CV(64,128;4c2s)-BN-LRec-CV(128,256;4c2s) -BN-LRec-CV(256,512;4c2s)-BN-LRec
- Autoregressive model for C-ARGAN: 2 layer LSTM, hidden size:128
- Autoregressive model for S-ARGAN: 1 layer LSTM, hidden size:512
- Autoregressive model for PARGAN: 1 layer LSTM, hidden size:512
- Patch-wise discriminator for PARGAN: CV(512,1;1c1s)

Network architecture for STL-10:

- Generator: FC(100,512*3*3)-BN-ReLU-DC(512,256;4c2s)-BN-ReLU-DC(256,256;4c2s)-BN-ReLU-DC(256,128;4c2s)-BN-ReLU-DC(128,3;4c2s)-Tanh
- Feature Learner: CV(3,64;4c2s)-BN-LRec-CV(64,128;4c2s)-BN-LRec-CV(128,256;4c2s) -BN-LRec-CV(256,512;4c2s)-BN-LRec
- Autoregressive model for C-ARGAN: 2 layer LSTM, hidden size:128
- Autoregressive model for S-ARGAN: 1 layer LSTM, hidden size:512
- Autoregressive model for PARGAN: 1 layer LSTM, hidden size:512
- Patch-wise discriminator for PARGAN: CV(512,1;1c1s)

FC(100,512) denotes fully connected layer from 100 to 512 dimensions, BN is batch-normalization, DC(512,256;4c2s) is transposed convolution with input channels 512 and output channels 256, while 4c2s denotes filter size 4 and stride 2 (fractional stride in this case). CV is convolution layer and LRec is leaky ReLU.

