# OpenReview forum: "Autoregressive Generative Adversarial Networks"
_ICLR.cc/2018/Conference — Invite to Workshop Track_

### Official Review · AnonReviewer3 · 2017-11-26
**interesting gan architecture, evaluations limited**

**Rating:** 5
**Confidence:** 4

**Review:**

This paper proposes a new GAN model whereby the discriminator (rather than being a binary classifier) consists of an encoding network followed by an autoregressive model on the encoded features. The discriminator is trained to maximize the probability of the true data and minimize the probability of the generated samples.  The authors also propose a version that combines this autoregressive discriminator with a patchGAN discriminator. The authors train this model on cifar10 and stl10 and show reasonable generations and inception scores, comparing the latter with existing approaches.

Pros: This discriminator architecture is well motivated, intuitive and novel. the samples are good (though not better than existing approaches as far as I can tell). The paper is also well written and easy to read.

Cons: As is commonly the case with GAN models, it is difficult to assess the advantage of this approach over exiting techniques. The samples generated form this model look fine, but not better than existing samples. The inception scores are ok, but seem to be outperformed by other models (though this shouldn't necessarily be taken as a critique of the approach presented here as inception scores are an approximation to what we care about and we should not be trying to tune models for better inception scores).

Detailed comments:
- In terms of experiments, I think think paper is missing the following: (1) An additional dataset -- cifar and stl10 are very similar, a face dataset for example would be good to see and is commonly used in GAN papers. (2) the authors claim their method is stable, so it would be good to see quantitative results backing this claim, i.e. sweeps over hyper-parameters / encoding/generator architectures with evaluations for different settings.
- the idea of having some form of  recurrent (either over channels of spatially)  processing in the discriminator seems more general that the specific proposal given here. Could the authors say a bit more about what they think the effects of adding recurrence in the discriminator vs optimizing the likelihood of the features under the autoregressive model?

Ultimately, the approach is interesting but there is not enough empirical evaluations.

---

> ### Author Response · Authors · 2018-01-05
> **Response to AnonReviewer3**
>
> Thanks for your review and feedback. (1) We have included CelebA dataset at 64x64 resolution with SW-ARGAN objective. (2) We did not claim stability over various architectures or hyperparameters. For setting that we mention in the paper, the method works well without mode collapse or training instability. Also our model is fairly simple and does not use any trick (like in improvedGAN) to improve performance. For your (3) point, our autoregressive modeling can be also modeled with a CNN similar to PixelCNN, so it is not a specific proposal about recurrent modeling. One possible benefit of using autoregression instead of recurrent discriminator is that it takes more bits of information from the objective (similar to EBGAN) instead of single score from the last time step (real/fake score).

---

### Official Review · AnonReviewer2 · 2017-11-27
**Autoregression in feature space**

**Rating:** 3
**Confidence:** 5

**Review:**

This work attempts to improve the global consistency of samples generated by generative adversarial networks by replacing the discriminator with an autoregressive model in an encoded feature space. The log likelihood of the classification model is then replaced with the log likelihood of the feature space autoregressive model. It's not clear what can be said with respect to the convergence properties of this class of models, and this is not discussed.

The method is quite similar in spirit to Denoising Feature Matching of Warde-Farley & Bengio (2017), as both estimate a density model in feature space -- this method via a constrained autoregressive model and DFM via an estimator of the score function, although DFM was used in conjunction with the standard criterion whereas this method replaces it. This is certainly worth mentioning and discussing. In particular the section in Warde-Farley & Bengio regarding the feature space transformation of the data density seems quite relevant in this work.

Unfortunately the only quantitative measurements reporter are Inception scores, which is known to be a poor measure (and the scores presented are not particularly high, either); Frechet Inception distance or log likelihood estimates via AIS on some dataset would be more convincing. On the plus side, the authors report an average over Inception scores for multiple runs. On the other hand, it sounds as though the stopping criterion was still qualitative.

---

> ### Author Response · Authors · 2017-12-25
> **Response to AnonReviewer2**
>
> Thanks for your review and feedback. We will include Denoising Feature Matching to our paper and make a clear comparison. Even though Denoising Feature Matching uses density estimation in the latent space there are major differences which makes learning dynamic of our model totally different then theirs. (i) As you mentioned their method is complementary to GAN objective while our method can be learned standalone. (ii) More importantly their discriminator (encoder + classifier) are trained as in original GAN objective which means that features learned from the data distribution are based on classifier's feedback not on density model's. This crucial difference make both works different than one another. (iii) In our model feature co-occurrences is modeled explicitly. (iv) Motivation for both works are totally different.
>
> Unfortunately, we could not include a second score (FID) into the revision due to time limitations.

---

### Official Review · AnonReviewer1 · 2017-11-28
**Interesting idea. Insufficient empirical support.**

**Rating:** 5
**Confidence:** 5

**Review:**

This paper proposes an alternative GAN formulation that replaces the standard binary classification task in the discriminator with a autoregressive model that attempts to capture discriminative feature dependencies on the true data samples.

Summary assessment:
The paper presents a novel perspective on GANs and an interesting conjecture regarding the failure of GANs to capture global consistency. However the experiments do not directly support this conjecture. In addition, both qualitative and quantitative results to not provide significant evidence of the value of this technique over and above the establish methods in the literature.

The central motivation of the method proposed in the paper, is a conjecture that the lack of global consistency in GAN-generated samples is due to the binary classification formulation of the discriminator. While this is an interesting conjecture, I am somewhat unconvinced that this is indeed the cause of the problem. First, I would argue that other high-performing auto-regressive models such as PixelRNN and PixelCNN also seem to lack global consistency. This observation would seem to violate this conjecture. More importantly, the paper does not show any direct empirical evidence in support of this conjecture.

The authors make a very interesting observation in their description of the proposed approach. In discussing an initial variant of the model (Eqns. (5) and (6) and text immediately below), the authors state that attempting to maximize the negative log likelihood of the auto-regressive modelling the generated samples results in unstable training. I would like to see more discussion of this point as it could have some bearing on the difficulty of GAN to model sequential data in general. Does the failure occurs because the auto-regressive discriminator is able to "overfit" the generated samples?

As a result of the observed failure of the formulation given in Eqns. (5) and (6), the authors propose an alternative formulation that explicitly removes the negative likelihood maximization for generated samples. As a result the only objective for the auto-regressive model is an attempt to maximize the log-likelihood of the true data. The authors suggest that this should be sufficient to provide a reliable training signal for the generator. It would be useful if the authors showed a representation of these features (perhaps via T-SNE) for both true data and generated samples.

Empirical results:
The authors experiments show samples (qualitative comparison) and inception scores (quantitative comparison) for 3 variants of the proposed model and compare these to methods in the literature. The comparisons show the proposed model preforms well, but does not exceed the performance of many of the existing methods in the literature.

Also, I fail to observe significantly more global consistency for these samples compared to samples of other SOTA GAN models in the literature. Again, there is no attempt made by the authors to make this direct comparison of global consistency either qualitatively or quantitatively.

Minor comment:
I did not see where PARGAN was defined. Was this the combination of the Auto-regressive GAN with Patch-GAN?

---

> ### Author Response · Authors · 2017-12-26
> **Response to AnonReviewer1**
>
> Thanks for your review and feedback. PixelRNN and PixelCNN are pixel prediction methods. Since adjacent pixels in images are highly correlated and pixel values can be captured by local image statistics, their model might be using most of their capacity to model local information instead of global. However in our case, autoregressive modeling is in latent space and adjacent features values are less correlated than adjacent pixel values. Also high level abstract representation are globally related such as co-occurrence of different object parts in a scene. As a result, PixelRNN's and PixelCNN's lack of global consistency is not necessarily about autoregressive modeling but more about their pixel level modeling.
>
> Maximizing Eqns. (5) by updating "R" parameters are unstable because the second term in the equation is unbounded. We mentioned this in paper but did not go into detail. When the second term is unbounded gradient of it gets way bigger than the first term's gradient. As a result optimization cares mostly about the second term which means it  decreases probability of generated feature while avoiding to increase probability of real features. We see this phenomenon exactly in our experiment. After some iterations the error of the first term starts to increase because the gradients cares about the second term. We tried a simple method to overcome this by using margin loss, as in EBGAN, in the second term which makes the second term bounded. However this trick did not provide better results than by simple discarding the second term from R's objective. We do not think it has something to do with sequential data but being unbounded.
>
> "The authors suggest that this should be sufficient to provide a reliable training signal for the generator".  Empirical evidence, both qualitative and quantitative, shows that this is a reliable training signal for the generator. Even though, the auto-regressor is not adversarial the encoder is adversarial which satisfies the distinguishability of real and fake samples. Our intuition is that as the auto-regressor fits on only real features it discovers feature co-occurrence statistics in real data distribution. Repelling from fake data distribution is not necessary since it is already satisfied by the encoder. As mentioned previously, margin loss in the second term is not better than discarding it totally.
>
> For Empirical results: We see that we emphasized global inconsistency a lot in the paper however it is just an observation about what is lacking in the current GAN models and why it might be happening. Generic theme of our model is learning a generative model by using feature co-occurrence statistics in real data distribution which is not found in generative distribution. Our C-ARGAN and S-ARGAN can learn both spatial layout and feature layout. Even though our model is not better than other GAN models, its competitive on DCGAN architecture and can be further improved with more advanced autoregressive modeling as we mentioned in the conclusion section.
>
> Sorry for PARGAN confusion. It is simply summation of the Auto-regressive GAN with Patch-GAN without any hyperparameters. It will be included into the revision.

---

### Decision · Program_Chairs · 2018-01-29
**ICLR 2018 Conference Acceptance Decision**

**Decision:**

Invite to Workshop Track

**Comment:**

The reviewers (all experts in this area) appreciated the novelty of the idea, though they felt that the experimental results (samples and Inception scores) did not provide convincing evidence value of this method over already established techniques. The authors responded to the concerns but were not able to address the issue of evaluation due to time constraints. The idea is likely sound but evaluation does not meet the bar, it may make a good contribution as a workshop paper.